# TADI: Topic-aware Attention and Powerful Dual-encoder Interaction for Recall in News Recommendation

**Junxiang Jiang**

jx.jiang@outlook.com

## Abstract

News recommendation is one of the widest commercialization in natural language processing research area, which aims to recommend news according to user interests. News recall plays an important role in news recommendation. It is to recall candidates from a very large news database. Recent researches of news recall mostly adopt dual-encoder architecture as it provides a much faster recall scheme, and they encode each word equally. However, these works remain two challenges: irrelevant word distraction and weak dual-encoder interaction. Therefore, we propose a model **T**opic-aware **A**ttention and powerful **D**ual-encoder **I**nteraction for recall in news recommendation (TADI). To avoid irrelevant word distraction, TADI designs a **T**opic-aware **A**ttention (TA) which weights words according to news topics. To enhance dual-encoder interaction, TADI provides a cheap yet powerful interaction module, namely **D**ual-encoder **I**nteraction (DI). DI helps dual encoders interact powerfully based on two auxiliary targets. After performance comparisons between TADI and state-of-the-arts in a series of experiments, we verify the effectiveness of TADI.

## 1 Introduction

News recommendation is one of the widest commercialization in natural language processing research area, which feeds rich and suitable news to users based on their interests. Currently, news recommendation is generally used on online news websites, movie review websites and etc (such as MSN News), it thus has become an useful tools to provide masses of custom information in one go. Generally, recall and ranking are two main steps of news recommendation (Wu et al., 2022). The first one is to recall candidates from a very large news database, while the second one is to rank news candidates for display. News recall determines the room of recommendation performance, and thus this paper discusses about it.

Researches of news recall (Wu et al., 2022) (or new candidate generation (Covington et al., 2016) or news retrieve (Wang et al., 2023)) can be categorized into feature-based and content-based models. Feature-based models focus on feature interaction modeling such as YoutubeNet (Covington et al., 2016) and Pinnersage (Pal et al., 2020). Feature-based models require to summarize features by manual text mining and thus they inevitably lose useful information. With the development of content understanding technology, researchers incorporate content understanding with feature interaction, that is content-based models, such as (Okura et al., 2017). Different from manual text mining of feature-based models, content-based models directly learn representations of user and news by modeling news contents. However, content-based models ignore irrelevant word distraction problem. Every word is encoded equally, that is why irrelevant words would bring side-effect for news recommendation. For example, football fans are more interested in the news "Lionel Messi comes back Spain for taking a holiday" than tourists, but only two words "Lionel Messi" in the title are related.

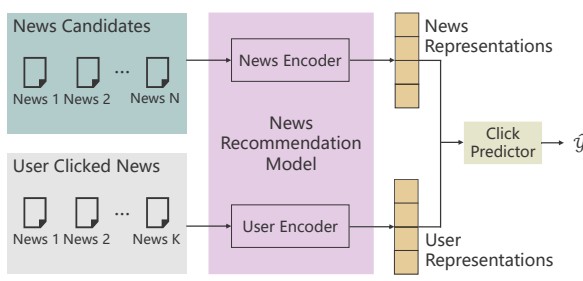

Figure 1: The typical dual-encoder model architecture (Mao et al., 2022).

To recall news candidates, researches mostly rely on dual-encoder architecture which is shown in Figure 1. Dual-encoder architecture is able to serve efficiently in real time for a large scale of news. Because it encodes user and news inde-

pendently, and solves the top-k nearest neighbor search problem in sublinear complexity, by converting the problem to **M**aximum-**I**nner-**P**roduct **S**earch (MIPS) (Yao et al., 2021). However, there exists a classical challenge of dual-encoder architecture (Khattab and Zaharia, 2020), that is weak dual-encoder interaction. Specifically, the click predictor (e.g., dot product), which unfortunately is the only interation between dual encoders in that architecture. Weak interaction makes one encoder difficultly utilize the information of the other encoder. Therefore, model underestimates actual correlation between dual encoders, resulting in severe performance degradation.

To response to the aforementioned challenges, we propose a news recall model, namely, **T**opic-aware **A**ttention and powerful **D**ual-encoder **I**nteraction for recall in news recommendation (TADI)[1]. First, we design the **T**opic-aware **A**ttention (TA) mechanism to avoid irrelevant word distraction. Because news topic is one of the most important interest indicators which directly reflects preference of potential target users. So it is reasonable to weight words by using TA. Secondly, TADI involves the **D**ual-encoder **I**nteraction (DI) module which helps dual encoders interact more powerful than typical dual-encoder models. In detail, DI provides two auxiliary targets to enhance dual encoders interaction on training, without changing the mechanism of online news recall in dual-encoder architecture. Therefore, TADI can leverage the efficiency of dual-encoder architecture on news recall while simultaneously gaining the ability to more powerful interaction. Afterwards, the effectiveness of TADI is verified by conducting a series of experiments on benchmark dataset MIND (Wu et al., 2020).

In summary, our contributions are four-fold: (1) We propose a news recall model TADI. (2) We design the topic-aware attention TA to avoid the distraction of irrelevant words. (3) We propose the dual-encoder interaction DI to enhance dual-encoder interaction. (4) Extensive experiments are conducted on the benchmark datasets, which demonstrate the effectiveness of TADI.

## 2 Related Works

Researches of news recall have rather matured works, mainly could be divided into fearture-

based models and content-based models. Furthermore, we introduce researches of news ranking because its technologies are available for news recall.

### 2.1 News Recall.

Feature-based models focus on feature interaction modeling, and they are usually utilized in product recommendation and movie recommendation. In our common live, YoutubeNet and Pinnersage are well known featured-based baselines in news recall (Wu et al., 2022). YoutubeNet uses the average of clicked news embeddings for recall. Pinnersage recall items based on hierarchical clustering. However, difficulty of effective content mining lead to information loss, which limits performances of feature-based models.

In contrast to feature-based models, content-based models pay attention to content understanding modeling. Most content-based recommendation models (Wu et al., 2019b) learn user representation from sequential user clicked news, and learn news representations from news candidates. Besides, regarding to description of diverse and multi-prained user, a few researchers find that a series of user interrest representation are more suitable than a single one. Authors of HieRec (Qi et al., 2021) research user interest reporesentations more deeply. They split user interest into category, sub-category and overall, so that to learn multiple representations for a user. Compared with single representation models, multiple representation models achieve better performance, but consume times of computing resources on click prediction. (Yu et al., 2022) aims to improve both the effectiveness and the efficiency of pre-trained language models for news recommendation, and finally achieves significant performance improvement. But it consumes more computing resources by training M+1 teacher models and distilling twice to get 2 student models.

We review many researches respectively on two main branches of news recall: feature-based and content-based models. However, irrelevant word distraction might impact model with learning confusion. TADI avoids the problem by utilizing news topic. Many researches (Qi et al., 2021; Wu et al., 2019c) involves news topic in modeling, but they seldom aware the distraction and take action to solve it. Furthermore, weak interaction makes one encoder difficultly utilize the information of the other encoder. TADI exploit powerful interaction

[1]The source code of TADI is available at `https://github.com/jx-jiang01/tadi`

between dual encoders for information utilization.

## 2.2 News Ranking.

Researches of news ranking can be also categorized into feature-based and content-based models. FM and DeepFM are well known featured-based baselines in news ranking. FM models second order feature interactions, while DeepFM models higher order feature interactions. Recently, researchers (Kang and McAuley, 2018; Sun et al., 2019) additionally model user sequential behaviors for performance improvement, e.g., SDM (Lv et al., 2019) respectively models long-term and short-term user sequential behaviors. Turn to content-based models, MINE (Li et al., 2022) makes the number of user insterest representaions controlable by tunable hyper-parameter. And the model achieve best performance when hyperparameter at 32. (Mao et al., 2022) enriches the semantics of users and news by building user and news graphs.

## 3 Problem Formulation

In this section, we give main notations and define the news recall problem. Features of news and user are the same as previous works (Qi et al., 2021). First, for news candidate defination, we make it represented by four type features: title, category, sub-category and title entity. The news title $t^n$ is a word sequence. Denote the category and sub-category of the news are $c^n$ and $s^n$. Denote the title entity as $d^n$ where consists of entities. Secondly, we assume a user has $N$ historical clicked news, and structure of each historical news representation is the same as news candidate. Denote titles as $\mathcal{T} = [t_1^u, t_2^u, ..., t_N^u]$, categories as $\mathcal{C} = [c_1^u, c_2^u, ..., c_N^u]$, sub-categories as $\mathcal{S} = [s_1^u, s_2^u, ..., s_N^u]$ and title entities as $\mathcal{D} = [d_1^u, d_2^u, ..., d_N^u]$. The target of news recall is to learn the mapping from users to the most relevant news. Technically, the target is to minimize the gap between the ground truth $y$ and the predicted label $\hat{y}$ via optimizing model parameters.

## 4 The Proposed Model

TADI is divided into four modules, i.e., user encoder, news encoder, predictor, and dual-encoder interaction, which is shown in Figure 2. The user encoder and the news encoder respectively generate embeddings of user and news. The predictor calculates dot product between embeddings of

user and news, in order to predict click probability. The dual-encoder interaction provides a capability which helps dual encoders interact more powerful.

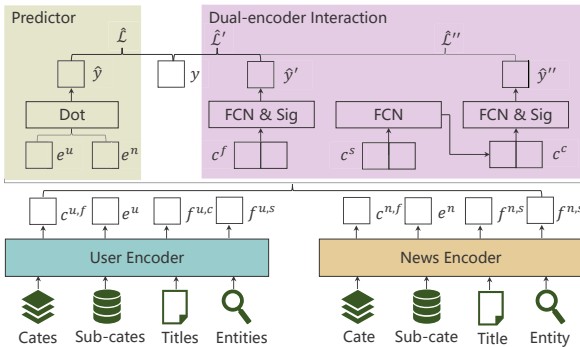

Figure 2: The overview of TADI architecture.

## 4.1 Basic Components

Before discussing the main modules of TADI, we introduce basic components at first, they are feature encoder, aggregation attention and topic-aware attention.

**Feature Encoder.** The purpose of feature encoder is to transform feature into dense embedding. Each news is represented by title, category, sub-category and title entity. First of all, similar to previous works (Qi et al., 2021), we adopt pre-trained models such as word2vec (Mikolov et al., 2013) and BERT to map word tokens of titles into dense embeddings. In experiment, we discuss advantages of word2vec and BERT in detail, for title encoding. For clarity, we name this type of feature encoder as title encoder. Secondly, category and sub-category are embeded by using GloVe (Pennington et al., 2014). Thirdly, entity embedding is learnt from knowledge graphs which is provided by datasets. Fourth, a shallow transformer encoder (2-layer and 4-head) are used to learn feature correlation of title and title entity. Fifthly, only for user encoding, all type embeddings are processed by shallow transformer encoders so that to learn cross-news information.

**Aggregation Attention.** Aggregation attention is used to integrate embeddings by using a query, a set of keys and values. The query and keys are used to calculate attention weights which measure the importance of values, and then the weighted sum of values is output. Suppose input is $\mathcal{X} = [x_1, x_2, ..., x_M]$ where $x_M \in \mathbb{R}^{d_x}$ and $d_x$ is embedding dimension, and $M$ is the number of embeddings which need to be aggregated. Inspired by the attention used in Poly-encoder (Humeau

et al., 2020), aggregation attention is designed in the same manner. The query is a trainable vector $q^a \in \mathbb{R}^{d_q}$. Keys $\mathcal{K}^a = [k_1^a, k_1^a, ..., k_M^a]$ where $k_M^a \in \mathbb{R}^{d_q}$, they are the output of a **Fully Connected Network (FCN)** whose input is $\mathcal{X}$, and values are the input itself $\mathcal{X}$. The attention weight is $\mathcal{A}^a = [\alpha_1^a, \alpha_2^a, ..., \alpha_M^a]$ where $\alpha_M^a \in \mathbb{R}$. In summary, we give the mathematical formula of aggregation attention as follows:

$$v^a = \sum_{i=1}^{M} \alpha_i^a x_i, \ \alpha^a = \mathrm{softmax}(q^{aT}\mathcal{K}) \quad (1)$$

**Topic-aware Attention.** Topic-aware attention aims to intergrate word embeddings by using topics, which getting rid of irrelevant word distraction. First, a topic embedding and word embeddings are used to generate a query and a series of key-value pairs. In detail, we map news topic embeding to $d$-dimension query $q^t \in \mathbb{R}^{d_t}$ by a FCN. And then we respectively map word embedings to keys $\mathcal{K}^t = [k_1^t, k_2^t, ..., k_M^t]$ where $k_M^t \in \mathbb{R}^{d_t}$ and values $\mathcal{V}^t = [v_1^t, v_2^t, ..., v_M^t]$ where $v_M \in \mathbb{R}^{d_t}$ by two FCNs. Secondly, we obtain attention weights $\alpha^t = [\alpha_1^t, \alpha_2^t, ..., \alpha_M^t]$ where $\alpha_M^t \in \mathbb{R}$. We scale down the dot product of $q^t$ and $\mathcal{K}^t$ by the square root of length $d_t$, and then normalize it to the attention weights by using softmax function. Thirdly, we aggregate $\mathcal{V}^t$ by using attention weights. The mathematical formula is below:

$$a^t = \sum_{i=1}^{M} \alpha_i^t v_i, \ \alpha^t = \mathrm{softmax}(\frac{q^{tT}\mathcal{K}^t}{\sqrt{d_t}}) \quad (2)$$

### 4.2 User Encoder

User Encoder is used to learn a user embedding from historical clicked news. The architecture of user encoder is shown in Figure 3, we introduce the main procedure below:

**Feature Aggregation.** Titles, categories, sub-categories and title entities of historical clicked news are transformed into dense embeddings $\mathcal{E}^{u,t}$, $\mathcal{E}^{u,c}$, $\mathcal{E}^{u,s}$ and $\mathcal{E}^{u,e}$ by using feature encoder. The above embeddings of historical clicked news need to be aggregated for unified embeddings obtaining. The aggregation operation is divided into two types according to feature type. First, category and sub-category aggregation. Categories and sub-categories embeddings of historical clicked news are respectively integrated into two embeddings $g^{u,c}$ and $g^{u,s}$ by using the aggregation attention. Secondly, title and title entity aggregation. Since

each news has several words and entities of the title, the aggregation module uses the aggregation attention to integrate embeddings of words and entities into $\mathcal{G}^{u,t}$ and $\mathcal{G}^{u,e}$. By doing so, we obtain title embedding and title entity embedding of each news. And then we use the aggregation attention twice to integrate title embeddings and title entity embeddings of historical clicked news for unified embeddings $g^{u,t}$ and $g^{u,e}$ obtaining.

**Topic-aware Encoding.** Models would be distracted by irrelevant words because such words are treated fairly with relevant words in feature encoding. The distration would be sevious especially when few related words in long title are avaiable to predict target. Therefore, user interest understanding by fair title encoding is not enough. As a supplement, model could utilize category and sub-category of news to identify relevant words in title. To do this, we use topic-aware attention to pay more attention to the topic correlated information.

**User Aggregation.** The target of user encoder is to learn a unified user embedding from historical clicked news. Therefore, all aggregated embeddings $g^{u,t}$, $e^{u,c}$, $f^{u,c}$, $e^{u,s}$, $f^{u,s}$ and $g^{u,e}$ are concatenated at first, then we map it to the user embedding $e^u$ via a FCN.

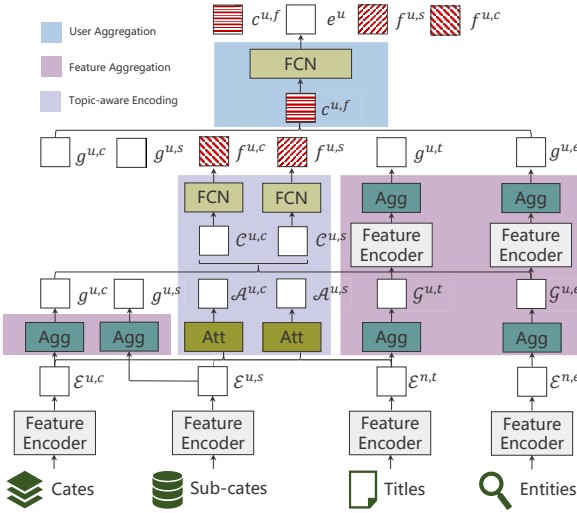

Figure 3: The architecture of user encoder.

### 4.3 News Encoder

The target of news encoder is to represent news by embeding learning. The architecture of news encoder is shown in Figure 4. In contrast to user encoder, the input of news encoder is only one news, which reduces a few aggregation operations. The procedure is similar to user encoder, so we

give a brief introduction below. First of all, title, category, sub-category and entity sequence of a news candidate are transformed into dense embeddings $\mathcal{E}^{n,t}$, $e^{n,c}$, $e^{n,s}$ and $\mathcal{E}^{n,e}$ by using feature encoder. Secondly, because the aforementioned embeddings of title and title entity are token-wise, so they are aggregated and we obtain $g^{n,t}$ and $g^{n,e}$. Thirdly, to avoid distraction of irrelevant words, we use category and sub-catogory to identify relevant information with topic-aware attention. As a result, we get category- and sub-catogory-wise embeddings $a^{n,c}$ and $a^{n,s}$. Finally, we integrate all embeddings and obtain news embedding $e^n$ by using a FCN.

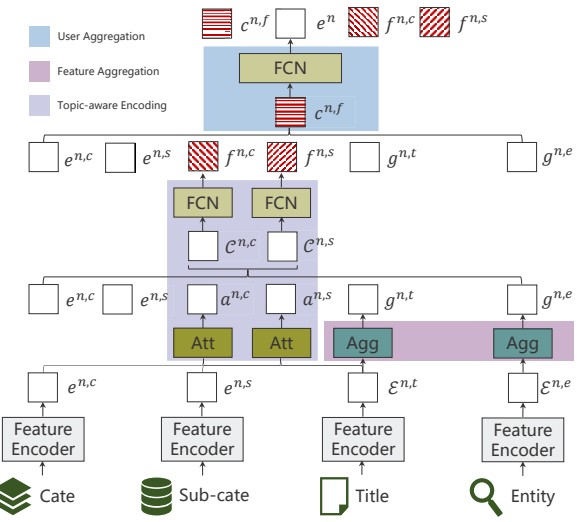

Figure 4: The architecture of news encoder.

## 4.4 Dual-encoder Interaction

In order to make dual encoders interact more powerfully, dual-encoder interaction module provides two auxiliary targets by utilizing some layer outputs of dual encoders. These two auxiliary targets are only used on training, so they do not change the mechanism of online news recall in dual-encoder architecture. The first one is powerful interaction (PI) target. It is to get more powerful interaction than only using dot product, by making use of top-level embeddings $c^{u,f}$ and $c^{n,f}$ to predict target. The formula is below. First of all, concatenating top-level embeddings and get the concatenated embedding $c^f$. Then, a FCN and a sigmoid function are used to predict labels $\hat{y}'$.

$$
\begin{aligned}
c^f &= \mathrm{concat}(c^{u,f}, c^{n,f}) \\
\hat{y}' &= \mathrm{sigmoid}(\mathrm{FCN}(c^f))
\end{aligned}
\tag{3}
$$

The second one is earlier interaction (EI) target, which aims to help model interact earlier, which is to predict target by using category- and sub-category-wise aggregated embeddings. To utilize hierarchical information between category and sub-category, we design a block. Specifically, the block first uses a FCN to process sub-category-wise concatenated embeddings $c^s$. Then, the block concatenates the above output with category-wise embeddings ($f^{u,c}$ and $f^{n,c}$). After processing by a FCN, we get $c^c$. Finally, $c^c$ is used to predict labels $\hat{y}''$ via a FCN and a sigmoid function. The formula is below:

$$
\begin{aligned}
c^s &= \mathrm{concat}(f^{u,s}, f^{n,s}) \\
c^c &= \mathrm{concat}(f^{u,c}, f^{n,c}, \mathrm{FCN}(c^s)) \\
\hat{y}'' &= \mathrm{sigmoid}(\mathrm{FCN}(c^c))
\end{aligned}
\tag{4}
$$

## 4.5 Optimization

The loss function $\mathcal{L}$ of TADI is divided into three parts: $\hat{\mathcal{L}}$, $\hat{\mathcal{L}}'$ and $\hat{\mathcal{L}}''$. $\hat{\mathcal{L}}$ is the loss between the predicted label $\hat{y}$ and the ground truth $y$, while $\hat{\mathcal{L}}'$ and $\hat{\mathcal{L}}''$ are losses of two auxiliary targets. Technically, they measure differences between their predicted labels ($\hat{y}'$ and $\hat{y}''$) and the ground truth. The mathematical formula of $\mathcal{L}$ is below:

$$
\begin{aligned}
\mathcal{L} = a\hat{\mathcal{L}} + b\hat{\mathcal{L}}' + (1 - a - b)\hat{\mathcal{L}}'' \\
where \quad a > 0, \ b > 0, \ 0 < a + b < 1
\end{aligned}
\tag{5}
$$

where $a$ and $b$ are hyper-parameters, they are set to 0.8 and 0.1 in expriments. Following previous works (Qi et al., 2021), $\hat{\mathcal{L}}$ utilizes **N**oise **C**ontrastive **E**stimation (NCE) loss. Given the $i$-th positive sample (a clicked news) in a batch of dataset, we randomly select $K$ negative samples (non-clicked news) for it. The selection is from the same news impression which displayed to the user. The NCE loss requires positve sample which assigning higher score than negative one and it is formulated as:

$$
\hat{\mathcal{L}} = -\sum_{i=1}^{N_b} \log \frac{\exp(\hat{y}_i^+)}{\exp(\hat{y}_i^+) + \sum_{j=1}^{K} \exp(\hat{y}_{ij}^-)}
\tag{6}
$$

where $N_b$ is the batch size. $\hat{\mathcal{L}}'$ and $\hat{\mathcal{L}}''$ utilize **B**inary **C**ross **E**ntropy (BCE) loss. Take $\hat{\mathcal{L}}'$ as an example, the formula is below:

$$
\mathcal{L}' = -\frac{1}{N_b} \sum_{i=1}^{N_b} y_i \log \hat{y}_i' + (1 - y_i) \log(1 - \hat{y}_i')
\tag{7}
$$

# 5 Experiment

We now turn our attention to empirically testing TADI, and conduct expriment analyses are to verify the effectiveness of our work.

Table 1: Dataset Statistic.

|  | MIND-small | MIND-large |
|---|---|---|
| # News | 65,238 | 161,013 |
| # Categories | 18 | 20 |
| # Sub-categories | 270 | 294 |
| # Impressions | 230,117 | 15,777,377 |
| # Clicks | 347,727 | 24,155,470 |

## 5.1 Experiment Setup

Experiments are conducted on MIND which is a benchmark in real-world news recommendation. The dataset includes two versions: MIND-large and MIND-small. Table 1 illustrates dataset statistics. MIND-large contains more than 15 million impression logs generated by 1 million users, from which MIND-small randomly samples 50,000 users. An impression log includes the clicked news, non-clicked news and historical clicked news of the user before this impression. Besides, each news contains title, category, sub-category and title entity. Following previous works (Qi et al., 2021), we employ four ranking metrics, i.e., AUC, MRR, nDCG@5, and nDCG@10, for performance evaluation. The evaluation metrics in our experiments are used on both news ranking models and recall models, such as previous works (Wang et al., 2023; Khattab and Zaharia, 2020; Cen et al., 2020). On the purpose of utilizing experimental results from previous works (such as (Qi et al., 2021; Li et al., 2022; Wu et al., 2020)), our experiments apply the same. The test ste of MIND-large does not have labels, so the evaluation is on an online website[2]. Our experiments are conducted on 12 vCPU Intel(R) Xeon(R) Platinum 8255C CPU@2.50GHz, 43GB memory and GPU RTX 3090. We count the time consumption of model training on MIND-small: running one epoch respectively consumes about 28 minutes and 120 minutes when using GloVe and MiniLM as title encoder.

We utilize users' most recent 40 clicked news to learn user representations. From each news, we use NLTK to split a title into words, then select the first 30 words. For title entity, we select

[2]https://codalab.lisn.upsaclay.fr/competitions/420

the first 10 entities. To explore the influence of pre-trained models to title encoder, we adopt the 300-dimensional GloVe and MiniLM (MiniLM-12l-384d, a distilled BERT) (Wang et al., 2021) to initialize title encoder, because MiniLM can save more time consumption than BERT. Embeddings of category and sub-category are initialized by using GloVe, and they are unfixed during training. The $K$ of Eq. 6 is set to 4 during training, which means each positive news is paired with 4 negative news. We employ Adam (Kingma and Ba, 2015) as the optimization algorithm.

## 5.2 Compared Models

Considering characteristic of title encoder, we categorize models into W2V and BERT types. W2V-based and BERT-based models mean that using W2V (such as word2vec, GloVe) or BERT-like (such BERT, MiniLM) model to encode titles.

**W2V.** (1) DKN (Wang et al., 2018): It uses CNN to learn news representation, and a target-aware attention network to learn user representation. (2) NPA (Wu et al., 2019b): It learns news and user representations by considering user personality. (3) NAML (Wu et al., 2019a): It learns user and news representations by using multi-view learning, and it is the **S**tate-**O**f-**T**he-**A**rt (SOTA) of single representation models with GloVe. (4) LSTUR (An et al., 2019): It models both short-term and long-term user interests by using GRU networks and user ID embeddings. (5) NRMS (Wu et al., 2019d): It employs multi-head self-attentions to learn user and news representations; (6) HieRec (Qi et al., 2021): To represent a user, it learns an overall embedding, embeddings for each category and embeddings for each sub-category. HieRec costs about 300 times of time consumption than single representation model for news recall.

**BERT.** (1) BERT: (Bi et al., 2022) only uses BERT for recall in news recommendation. (2) LSTUR+BERT (Wu et al., 2021): It uses BERT as title encoder on LSTUR. (3) NRMS+BERT: NRMS uses BERT as title encoder, which is the SOTA of single representation models with BERT.

## 5.3 Experiment Analysis

In this section, we first analyze model performance. By doing so, we conduct an ablation analysis. Finally, extensive analyses illustrate the effect of embedding dimension and model performance on different title encoders.

Table 2: Performance Analysis. Baseline performances are provided by (Li et al., 2022; Qi et al., 2021; Wu et al., 2021; Zhang et al., 2021). **Bold** means the best performance, while underline means the best performance of baseline models. We repeated experiment TADI 3 times and reported average result with standard deviation.

| Feature | Model | MIND-small | | | | MIND-large | | | |
|---|---|---|---|---|---|---|---|---|---|
| | | AUC | MRR | nDCG@5 | nDCG@10 | AUC | MRR | nDCG@5 | nDCG@10 |
| Manual | LibFM | 59.74 | 26.33 | 27.95 | 34.29 | 61.85 | 29.45 | 31.45 | 37.13 |
| | DeepFM | 59.89 | 26.21 | 27.74 | 34.06 | 61.87 | 29.30 | 31.35 | 37.05 |
| W2V | DKN | 62.90 | 28.37 | 30.99 | 37.41 | 64.07 | 30.42 | 32.92 | 38.66 |
| | NPA | 64.65 | 30.01 | 33.14 | 39.47 | 65.92 | 32.07 | 34.72 | 40.37 |
| | NAML | 66.12 | 31.53 | 34.88 | 41.09 | 66.46 | 32.75 | 35.66 | 41.40 |
| | LSTUR | 65.87 | 30.78 | 33.95 | 40.15 | 67.08 | 32.36 | 35.15 | 40.93 |
| | NRMS | 65.63 | 30.96 | 34.13 | 40.52 | 67.66 | 33.25 | 36.28 | 41.98 |
| | HieRec | 67.95 | 32.87 | 36.36 | 42.53 | 69.03 | 33.89 | 37.08 | 43.01 |
| | TADI | **68.28** ± **0.07** | **33.05** ± **0.12** | **36.75** ± **0.15** | **42.91** ± **0.13** | **69.53** ± **0.11** | **34.35** ± **0.03** | **37.42** ± **0.03** | **43.12** ± **0.03** |
| BERT | BERT | 68.26 | 32.52 | 35.89 | 42.33 | - | - | - | - |
| | LSTUR+BERT | 68.28 | 32.58 | 35.99 | 42.32 | 69.49 | 34.72 | 37.97 | 43.70 |
| | NRMS+BERT | 68.60 | 32.97 | 36.55 | 42.78 | 69.50 | **34.75** | **37.99** | **43.72** |
| | TADI | **69.39** (± **0.17**) | **33.68** ± **0.24** | **37.55** ± **0.2** | **43.65** ± **0.21** | **70.00** ± **0.18** | 34.49 ± 0.13 | 37.69 ± 0.15 | 43.45 ± 0.14 |

### 5.3.1 Performance Analysis

We compare model performances to demonstrate effectiveness of our work, in the perspective of title encoders. Table 2 illustrates the performance of each model on MIND-small and MIND-large, from which we have following observations:

**W2V.** First of all, TADI is the best performance model, which verifies the effectiveness of our work. Secondly, performance gaps are large comparing TADI with single representation models both on MIND-small and MIND-large. From the comparisons, we find that TADI achieves significant improvement over baseline models, and the comparison results powerfully support the effectiveness of our work. Thirdly, TADI is better than multiple representation models no matter on performance or online speed. Performance gaps are smaller between TADI and HieRec than previous comparisons, but TADI is much faster than HieRec for news recall. Efficiency is the key feature when we considering TADI. The reason why TADI is able to achieve good efficiency is because DI only exists on model training, to help the model obtain interactive information and achieve better performance on news recall. Therefore, it does not add additional computing complexity to embedding inference and news recall. When measuring efficiency, news recall only considers the time consumption of rating because user embedding and news embedding can be inferred offline. The way that TADI recalls news is similar with basic dual-encoder models, that is calculating the

dot product of two embeddings. However, HieRec trains multiple embeddings for each user and each news. For example, in the MIND small dataset, each user and each news respectively have 289 embeddings (1 overall embedding, 18 category-wise embeddings, and 270 sub-category-wise embeddings) for rating. Therefore, when recalling news, the time consumption of HieRec is 289 times that of TADI.

Table 3: Ablation Analysis on MIND-small.

| Module | AUC | MRR | nDCG@5 | nDCG@10 |
|---|---|---|---|---|
| w/ W2V | 68.36 | 33.15 | 36.81 | 43.01 |
| w/o PI | 68.33 | 33.13 | 36.80 | 42.98 |
| w/o EI | 68.32 | 33.08 | 36.78 | 42.96 |
| w/o DI | 67.81 | 32.85 | 36.49 | 42.66 |
| w/o all | 67.23 | 32.45 | 36.11 | 42.30 |
| w/ BERT | 69.28 | 33.51 | 37.39 | 43.48 |
| w/o PI | 69.31 | 33.55 | 37.48 | 43.52 |
| w/o EI | 69.33 | 33.60 | 37.49 | 43.55 |
| w/o DI | 69.05 | 33.45 | 37.17 | 43.34 |
| w/o all | 68.42 | 32.78 | 36.38 | 42.67 |

**BERT.** First, similar to the previous analyses on W2V, TADI is better than baseline models with large performance gap. This observation demonstrates the effectiveness of TADI with BERT. Secondly, compared with using W2V to encode titles, TADI with BERT performs better. From the comparison, we find that it is worth to use BERT on title encoding, even thought it brings more computing complexity.

**Summary.** The proposed TADI outperforms

the SOTA whenever using W2V or BERT type models to encode title. Furthermore, large performance gaps between TADI and single representation models illustrate that TADI achieves significant improvement. Finally, TADI demonstrates that single representation models are competitive in contrast to multiple representation models.

### 5.3.2 Ablation Analysis

To understand the importance of TA and DI, we conduct an ablation analysis, as shown in Table 3. Observations are that: First of all, we verify the importance of DI. Performance degradates after removing DI, which reveals that DI is necessary for TADI. Secondly, we furtherly remove TA to analyze its effect. After removing TA, we observe that the performance of TADI further declines. Therefore, the effect of TA is enormous for TADI. Thirdly, we verify the importance of PI and EI targets in DI. After comparing their performances, we find that one of them for TADI is enough. Fourth, we combine Table 2 and Table 3, TADI without TA and DI is already outperforming most of the baselines. The reasons are that: First, title encoder uses strong component (transformer encoder) to learn cross information among history clicked news, while NAML ignores them and LSTUR adopts a weak component (GRU). Secondly, TADI concatenates shallow and deep information when integrating all information after feature encoder, while baselines don't. Therefore, TADI without TA and DI but also achieves better performance.

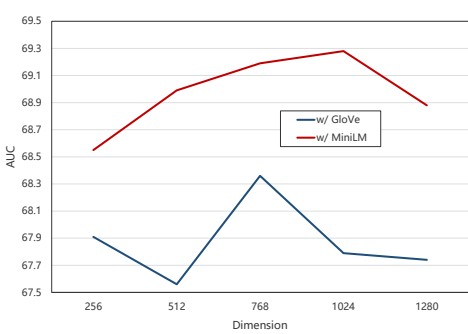

Figure 5: Dimension Analysis on MIND-small.

### 5.3.3 Embedding Dimension Analysis

We explore the optimal embedding dimension of TADI with GloVe or MiniLM on MIND-small, as shown in Figure 5. We observe that: First, TADI with GloVe achieves the optimal performance when the dimension is set to 768. In de-

tail, performance gaps between the optimal performance and the rest are larger than 0.4%. Secondly, different from using GloVe, TADI with MiniLM achieves the best performance when the dimension is set to 1024. Specifically, the performance is continously rising until the dimension reaches to 1024, and the performance declines when the dimension large than 1024. In summary, it is optimal to set the embedding dimension as 768 when TADI uses GloVe, and to set the embedding dimension as 1024 when TADI uses MiniLM.

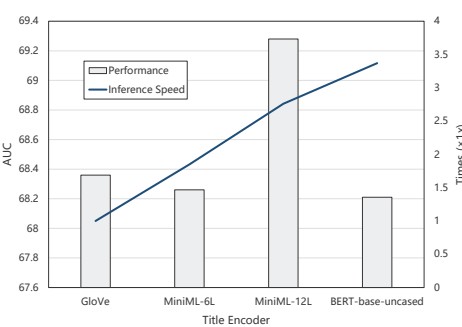

Figure 6: Title Encoder Analysis on MIND-small.

### 5.3.4 Title Encoder Analysis

We conduct an experiment to analyze the influence of title encoder. Previously, we have analyzed TADI performance by using GloVe and MiniLM as title encoder, but MiniLM is only one of distilled BERT versions. Besides, title encoders of a few baselines are different from them. Therefore, we compare the performance and inference speed of TADI by using more title encoders. From comparisons, title encoders are respectively set to GloVe, MiniLM-6L (MiniLM-6l-384d), MiniLM-12L (MiniLM-12l-384d) and BERT-base-uncased. From Figure 6, we observe: First, TADI with MiniLM-12L achieves the best performance. Because MiniLM-12L learns more useful information from pre-training than GloVe and MiniLM-6L. And the data scale of MIND-small might not fine-tune BERT-base-uncased well, which makes it perform worse. Secondly, the inference speed of TADI with GloVe is the fastest. With the increasing of model complexity, inference speed becomes slower and slower. In summary, we prefer to use GloVe or MiniLM-12L as title encoder in terms of performance and inference speed.

### 5.3.5 Topic-aware Attention Analysis

We conduct case studies to verify the effectiveness of TA. Five news are randomly selected and

Table 4: Case Studies of TA.

| | Title | Category | Top 5 Relevant Words (desc) | Sub-category | Top 5 Relevant Words (desc) |
|---|---|---|---|---|---|
| 1 | Senior Trump official embellished résumé, had face on fake Time cover | news | Senior, had, fake, Trump, embellished | news politics | résumé, embellished, fake, on, Trump |
| 2 | 2020 Ford Explorer launch hardly went according to plan, company admits | autos | Ford, admits, company, launch, Explorer | autos news | Explorer, 2020, hardly, company, plan |
| 3 | Mother of missing Florida girl charged; human remains found | news | girl, missing, human, remains, charged | news crime | missing, girl, Florida, charged, remains |
| 4 | Wealthy investors are bracing for a sharp stock sell-off in 2020 | finance | for, sell-off, investors, in, stock | markets | sell-off, investors, bracing, Wealthy, stock |
| 5 | Chrissy Teigen 's weekend was basically a double-header, plus more news | entertainment | double-header, Chrissy, Teigen, weekend, basically | entertainment celebrity | Teigen, Chrissy, double-header, plus, basically |

the news encoder is used to calculate the attention weights of their words. Words are sorted in descending order by the attention weight, and we list top five words in Table 4. From the table, we observe that: First of all, with topic-aware attention, both category and subcategory capture the correlative words. Top five words are generally correlative to the category or subcategory. Secondly, the information of category and subcategory capturing is complementary, which improves the performance of TADI. The above comparisons verify the effectiveness of TA.

Table 5: Quantitatively Analysis of Topic-aware Attention. The number within parentheses is the normalized rank. JJT means morphologically superlative adjective. VBD means verb, past tense. VBG means verb, present participle/gerund. OD means ordinal numeral. VBN means verb, past participle. DT means singular determiner. WDT means wh- determiner. BEZ means the word "is". DO means the word "do".

| | POS (Desc by the rank) |
|---|---|
| Top 5 | JJT (0.3512), VBD (0.3883), VBG (0.4351), OD (0.4447), VBZ (0.4478) |
| Last 5 | . (0.7167), DT (0.7177), WDT (0.7298), BEZ (0.7303), DO (0.7417) |

To quantitatively analyze the effectiveness of TA based on a whole dataset, we additionally provide Part-Of-Speech (POS) rank. We firstly use NLTK to tag POS of each title and rank POS by using topic attention weights (category-wise). Secondly, to normalize the rank, we divide the rank by the word size of the title. Finally, we select POS with frequency large than 50, and count their average ranks. By doing so, we list the top 5 POS and the last 5 POS in Table 5. We observe that TA pays more attention to informative words, for example, JJT is obviously more informative than BEZ, so TA is effective.

## 6 Conclusion

In this paper, we propose the model TADI for news recommendation. To avoid irrelevant word distraction, TADI designs the Topic-aware Attention (TA). TA uses news topic to weight words since news topic is one of important insterest indicators which directly reflects preference of potential target users. To make dual encoders interaction more powerful, TADI provides Dual-encoder Interaction (DI). DI helps dual-encoder interact more powerfully by providing two auxiliary targets. After conducting a series of experiments, the effectiveness of the proposed TADI is verified. Besides, extensive analyses demonstrate the robustness of the proposed model. In the future, we plan to achieve better performance by TADI optimization and data augmentation usage.

## Limitations

Although TADI achieves good performance, but it remains limitations. First, DI is inflexible. DI could not be directly utilized to other models. Secondly, lack of data cannot fully fine-tune pretrained feature encoder.

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

# A  Appendix

We response some questions.

**Q1:** TADI seems to overlook a crucial problem: the training and inference ranking objective is discrepant.

First of all, I agree with the viewpoint: "the training and inference ranking objective is discrepant". But we could not say it's a problem. Multi-target losses allow the main target to learn knowledge from other targets. By weighting losses of multi-targets through parameters, we can retain beneficial knowledge to the main target, thereby the main target could achieve better performance. Secondly, we agree that multi-task gradient conflict sometimes makes the main target achieve bad performance, but in our experiments, we find that the main target performs better. In experiment, we have considered the conflicts among losses and have tried some methods to alleviate conflicts, such as GradNorm [1] and PCGrad [2].These methods consume a lot of extra training time but contribute not much performance improvement. Considering the goal of our paper, we believe it is worth to have more discussion in a separate paper. Finally, it is correct on the point of view "the interaction performance and efficiency are always in a trade-off relation" when using user and news embeddings to recall news. Nonetheless, researchers think about to do something to help dual-encoder models better such as ColBERT [3]. To achieve better performance, TADI enhance interaction of dual-encoder models on training procedure. Therefore, TADI will not reduce the efficiency of the model when recalling news, as it still rates news by calculating the dot product of two embeddings.