# OpenReview forum: "TADI: Topic-aware Attention and Powerful Dual-encoder Interaction for Recall in News Recommendation"
_EMNLP/2023/Conference — EMNLP 2023 Findings_

### Official Review · Reviewer_fcXJ · 2023-07-26

**Soundness:** 4

**Excitement:**

4: Strong: This paper deepens the understanding of some phenomenon or lowers the barriers to an existing research direction.

**Missing References:**

None

**Paper Topic And Main Contributions:**

The paper introduces a model called TADI (Topic-aware Attention and powerful Dual-encoder Interaction) for news recommendation to improve recall. TADI addresses irrelevant word distraction by using Topic-aware Attention (TA), which weights words based on news topics, reflecting the preferences of potential target users. Additionally, TADI enhances the interaction between dual encoders with Dual-encoder Interaction (DI), providing two auxiliary targets. Through experiments, the effectiveness and robustness of TADI are verified and analyzed.






**Questions For The Authors:**

1. Can you used more datasets to verify your method?

**Reasons To Accept:**

1. The proposed studied in this paper is practical, important and interesting.

2. The proposed method is solid and reasonable in my opinion.

3. The experimental results are promising and convincing.

**Reasons To Reject:**

1. It would be better if more datasets are used for experiments. Currently, only two versions of MIND dataset is used.

2. There are a few writing flaws, such as "Table 2 and 3" and "Ablation Analysis on MIND-small".

**Reproducibility:**

4: Could mostly reproduce the results, but there may be some variation because of sample variance or minor variations in their interpretation of the protocol or method.

**Reviewer Confidence:**

4: Quite sure. I tried to check the important points carefully. It's unlikely, though conceivable, that I missed something that should affect my ratings.

**Typos Grammar Style And Presentation Improvements:**

There are a few writing flaws, such as "Table 2 and 3" and "Ablation Analysis on MIND-small".

---

> ### Author Rebuttal · Authors · 2023-08-29
>
> Thank you for your constructive comments and suggestions. In the following, we give point-by-point responses to your questions.
>
> >It would be better if more datasets are used for experiments. Currently, only two versions of MIND dataset is used.
>
> Experimentation on more datasets can achieve higher credibility, and we also hope to evaluate the performance of TADI on more datasets. However, there is currently no publicly available dataset like MIND. As with many previous works, we conducted experiments on two versions of MIND datasets and verified the performance of TADI.

---

### Official Review · Reviewer_Gomy · 2023-08-03

**Soundness:** 3

**Excitement:**

3: Ambivalent: It has merits (e.g., it reports state-of-the-art results, the idea is nice), but there are key weaknesses (e.g., it describes incremental work), and it can significantly benefit from another round of revision. However, I won't object to accepting it if my co-reviewers champion it.

**Paper Topic And Main Contributions:**

This paper studies news recommendation. The authors propose a model TADI that can obtain good performance via dual interaction and the interaction efficiency is high. The core contribution is the proposed topic-aware attention and dual-encoder interaction module, which can achieve effective embedding learning and interaction between news and user sides and keep good efficiency in the inference stage.

**Questions For The Authors:**

Could you provide inference time compared to the baselines to justify the efficiency of DI?

**Reasons To Accept:**

1. The writing is clear and easy to understand.

2. The proposed model is modularized making it easily adapted to larger content-based encoders, for example, recent LLM models.

3. The motivation is clear and the model design is rational.


**Reasons To Reject:**

1. The authors seem to overlook a crucial problem: the training and inference ranking objective is discrepant. Specifically, the training objective of is $a\mathcal{\hat{L}}+b\mathcal{\hat{L}'}+(1-a-b)\mathcal{\hat{L}''}$, whereas the inference target is only related to $\mathcal{\hat{L}}$. This training/inference discrepancy is not beneficial to model performance (even harmful from my empirical experience on score-oriented recommendation, but I keep neutral to this point). The major reason could be multi-task gradient conflict [1, 2]. That is, the multi-task objectives tend to be disentangled and conflicted, one objective cannot well represent the other. In this paper, $\mathcal{\hat{L}}$, $\mathcal{\hat{L}'}$ and $\mathcal{\hat{L}''}$ are jointly optimized in training, solely evaluating based on $\mathcal{\hat{L}}$ may be inferior. The author may consider gradient surgery [2] to alleviate this problem. Frustratingly, previous studies and industry practice in single/dual recommender encoders show that the interaction performance and efficiency are always in a trade-off relation.

2. Because this paper claims an efficient interaction module, efficiency comparisons with baselines in inference are very important and should be done, but we cannot see it in the experiment tables. The authors claim the overhead of dot product on embeddings is "1/300 in contrast to HieRec" in L500. I think this is not rigorous because the authors neither give detailed inference time comparisons nor give more comprehensive explanations, given the fact that dot-product on embeddings is just a very lightweight computation compared to previous FCN and Transformer computation.

3. Experiments three times on MIND-small and once on MIND-large is not statistically supportive. This is because MIND-small is literally small and always leads to high experiment variance. This may be the reason why the curve in Figure 5 fluctuates. I think more experiment trials like previous papers [3, 4] can better support the paper.

4. Topic-aware attention was widely studied in previous papers [3, 4, 5], which is somehow incremental.


[1] Gradient Surgery for Multi-Task Learning. Neurips 2020.

[2] Conflict-Averse Gradient Descent for Multi-task Learning. Neurips 2021.

[3] HieRec: Hierarchical User Interest Modeling for Personalized News Recommendation. ACL 2021.

[4] MINER: Multi-Interest Matching Network for News Recommendation. ACL 2022.

[5] Neural news recommendation with topic-aware news representation. ACL 2019.

**Reproducibility:**

2: Would be hard pressed to reproduce the results. The contribution depends on data that are simply not available outside the author's institution or consortium; not enough details are provided.

**Reviewer Confidence:**

4: Quite sure. I tried to check the important points carefully. It's unlikely, though conceivable, that I missed something that should affect my ratings.

**Typos Grammar Style And Presentation Improvements:**

L86: "To response the" -> "To response to the"

L380 Equation (5): "s.t." abbreviates "subject to" meaning the constraint term in an optimization problem, here "where" is more appropriate.

L506: "encode title" -> "encode titles"

L527: "further decline" -> "further declines"

A suggestion: the word "aux" should be "auxiliary" in formal paper writing.

---

> ### Author Rebuttal · Authors · 2023-08-29
>
> We'd like to thank the reviewer for detailed notes and feedback. We response your questions below.
>
> >The authors seem to overlook a crucial problem: the training and inference ranking objective is discrepant...
>
> First of all, I agree with your viewpoints: "the training and inference ranking objective is discrepant". But we could not say it's a problem. Multi-target losses allow the main target to learn knowledge from other targets. By weighting losses of multi-targets through parameters, we can retain beneficial knowledge to the main target, thereby the main target could achieve better performance. Secondly, we agree that multi-task gradient conflict sometimes makes the main target achieve bad performance, but in our experiments, we find that the main target performs better. In experiment, we have considered the conflicts among losses and have tried some methods to alleviate conflicts, such as GradNorm [1] and PCGrad [2].These methods consume a lot of extra training time but contribute not much performance improvement. Considering the goal of our paper, we believe it is worth to have more discussion in a separate paper. Finally, it is correct on the point of view "the interaction performance and efficiency are always in a trade-off relation" when using user and news embeddings to recall news. Nonetheless, researchers think about to do something to help dual-encoder models better such as ColBERT [3]. To achieve better performance, TADI enhance interaction of dual-encoder models on training procedure. Therefore, TADI will not reduce the efficiency of the model when recalling news, as it still rates news by calculating the dot product of two embeddings.
>
> >Because this paper claims an efficient interaction module, efficiency ...
>
> First of all, efficiency is the key feature when we considering TADI. The reason why TADI is able to achieve good efficiency is because DI only exists on model training, to help the model obtain interactive information and achieve better performance on news recall. Therefore, it does not add additional computing complexity to embedding inference and news recall. Secondly, we respond to the question "why TADI is more efficient than HieRec on news recall". When measuring efficiency, news recall only considers the time consumption of rating because user embedding and news embedding can be inferred offline. The way that TADI recalls news is similar with basic dual-encoder models, that is calculating the dot product of two embeddings (i.e., user and news embeddings). However, HieRec trains multiple embeddings for each user and each news. For example, in the MIND small dataset, each user and each news respectively have 289 embeddings (1 overall embedding, 18 category-wise embeddings, and 270 sub-category-wise embeddings) for rating. Therefore, when recalling news, the time consumption of HieRec is 289 times that of TADI.
>
> >Experiments three times on MIND-small and once on MIND-large is not statistically supportive ...
>
>  Once Experiments on MIND-large is because the scale of MIND-large dataset is very large, many methods have given up conducting experiments on this dataset, including the mentioned HieRec [4]. More experiments are for robustness purpose, therefore,  once experiments on MIND-large is acceptable which not impact our conclusion, and it also not impact the comparison between MIND-large and MIND-small. But we repeat experiment two times on MIND-large when using GloVe and BERT as  title encoder, to make our work more convincing. The average results with the standard deviation are reported as below. For hyper-parameter analysis, the mentioned MINER [5] is also performed on MIND-small, so we are reasonable to analyze hyper-parameters on MIND-small.
> |Feature|AUC|MRR|nDCG@5|nDCG@10|
> |----|----|----|----|----|
> |W2V|69.53 $\pm$ 0.11|34.35 $\pm$ 0.03|37.42 $\pm$ 0.03|43.12 $\pm$ 0.03|
> |BERT|70.00 $\pm$ 0.18|34.49 $\pm$ 0.13|37.69 $\pm$ 0.15|43.45 $\pm$ 0.14|
>
>
> >Topic-aware attention was widely studied in previous papers, which is somehow incremental.
>
> Admittedly, there is a lot of works that have involved news topics into modeling, but seldom works have utilized topic-aware attention to alleviate irrelevant word distraction.
>
> [1] GradNorm: Gradient Normalization for Adaptive Loss Balancing in Deep Multitask Networks
>
> [2] Gradient Surgery for Multi-Task Learning
>
> [3] Omar Khattab et al. ColBERT: Eficient and Effective Passage Search via Contextualized Late Interaction over BERT. SIGIR'20
>
> [4] HieRec: Hierarchical User Interest Modeling for Personalized News Recommendation. ACL 2021.
>
> [5] MINER: Multi-Interest Matching Network for News Recommendation. ACL 2022.

---

### Official Review · Reviewer_o5wg · 2023-08-04

**Typos Grammar Style And Presentation Improvements:** Change 'new recall' to 'news recall'
**Soundness:** 3

**Excitement:**

3: Ambivalent: It has merits (e.g., it reports state-of-the-art results, the idea is nice), but there are key weaknesses (e.g., it describes incremental work), and it can significantly benefit from another round of revision. However, I won't object to accepting it if my co-reviewers champion it.

**Missing References:**

Yang Yu, Fangzhao Wu, Chuhan Wu, Jingwei Yi, and Qi Liu. 2022. Tiny-NewsRec: Effective and Efficient PLM-based News Recommendation. In Proceedings of the 2022 Conference on Empirical Methods in Natural Language Processing, pages 5478–5489, Abu Dhabi, United Arab Emirates. Association for Computational Linguistics.

**Paper Topic And Main Contributions:**

This work proposes a news recommender system by designing topic-award attention and dual-encoder interaction modules. The proposed model was evaluated on the MIND datasets, showing the highest performance compared to baseline models.

**Questions For The Authors:**

Why do you use the term 'recall' in news recommendation? I'm confused by the term as the task conducted in the experiment is simply the news recommendation (ranking). Is there any reference mentioning the term in news recommendation field?

**Reasons To Accept:**

1. The proposed deep learning architecture achieves the highest performance compared to baseline models.
2. The paper is generally well written, and thus readers will be easy to fully understand the idea in the paper.

**Reasons To Reject:**

1. The paper does not include an important related work [1] while the performance of [1] on MIND-large dataset is higher than the numbers reported in this paper. Therefore, the paper should compare the proposed approach with the recent related work [1].

2. ~~Analysis on topic-aware attention was missing. The goal of topic-aware attention is to avoid irrelevant word distraction. However, in the experiment, the paper does not show whether the goal is indeed achieved or not.~~

[1] Yang Yu, Fangzhao Wu, Chuhan Wu, Jingwei Yi, and Qi Liu. 2022. Tiny-NewsRec: Effective and Efficient PLM-based News Recommendation. EMNLP'22.

**Reproducibility:**

3: Could reproduce the results with some difficulty. The settings of parameters are underspecified or subjectively determined; the training/evaluation data are not widely available.

**Reviewer Confidence:**

5: Positive that my evaluation is correct. I read the paper very carefully and I am very familiar with related work.

---

> ### Author Rebuttal · Authors · 2023-08-29
>
> We thank the reviewer for the constructive comments. We response reasons of rejection point by point.
>
> >The paper does not include an important related work [1] while the performance of [1] on MIND-large dataset is higher than the numbers reported in this paper. Therefore, the paper should compare the proposed approach with the recent related work [1].
>
> Thanks for your kind reminding of this newly published work[1]. It is worthy and not late for us to compare 2 papers, after then we will supplement it in the Related Works and the Appendix.
>
>
> We analyze [1] and our work as below:
>
> 1. We compare [1] with our work. The goal of [1] is to enhance news encoder to make performance improvement, by achieving that [1] proposed Tiny-NewsRec. In our paper, we noticed irrelevant word distraction and weak interaction might reduce recommendation performance, by alleviating them we proposed TADI. Honestly Tiny-NewsRec is 1% better than TADI on AUC. We do analysis and the reasons might that: (1) Tiny-NewsRec enhances news encoder with an additional dataset, so theoretically Tiny-NewsRec are supposed to be better. In this case, comparisons between TADI and Tiny-NewsRec seem inappropriate.  (2) In contrast to TADI which trains only one model, Tiny-NewsRec trains M+1 teacher models and distills twice to get 2 student models. Tiny-NewsRec gets more information by using a much more complex but effective manner. Therefore, Tiny-NewsRec and TADI have their own advantages and disadvantages, people could do option based on actual requirements.
> 2. From goal achievement perspective, we successfully verify the effectiveness of TADI by comparing with baselines and conducting ablation analyses, no matters if we compare TADI with Tiny-NewsRec or not.
>
> >Analysis on topic-aware attention was missing. The goal of topic-aware attention is to avoid irrelevant word distraction. However, in the experiment, the paper does not show whether the goal is indeed achieved or not.
>
> Sorry to confused you on the verification of topic-aware attention. Please allow us to clarify here. We conduct a case study to verify the effectiveness of TA. Five news are randomly selected and the news encoder is used to calculate the attention weights of their words. Words are sorted in descending order by the attention weight, and we list top five words in the below table. From the table, we observe that: First, with topic-aware attention, both category and subcategory capture the correlative words. Top five words are generally correlative to the category or subcategory. Secondly, the information of category and subcategory capturing is complementary, which improves the performance of TADI. The above comparisons verify the effectiveness of TA.
>
> |Title|Category|Top 5 relevant words (desc)|Sub-category|Top 5 relevant words (desc)|
> |----|----|----|----|----|
> |Senior Trump official embellished résumé, had face on fake Time cover|news|Senior, had, fake, Trump, embellished|news politics|résumé, embellished, fake, on, Trump|
> |2020 Ford Explorer launch hardly went according to plan, company admits|autos|Ford, admits, company, launch, Explorer|autos news|Explorer, 2020, hardly, company, plan|
> |Mother of missing Florida girl charged; human remains found|news|girl, missing, human, remains, charged|news crime|missing, girl, Florida, charged, remains|
> |Wealthy investors are bracing for a sharp stock sell-off in 2020|finance|for, sell-off, investors, in, stock|markets|sell-off, investors, bracing, Wealthy, stock|
> |Chrissy Teigen 's weekend was basically a double-header, plus more news|entertainment|double-header, Chrissy, Teigen, weekend, basically|entertainment celebrity|Teigen, Chrissy, double-header, plus, basically|
>
> >Why do you use the term 'recall' in news recommendation? I'm confused by the term as the task conducted in the experiment is simply the news recommendation (ranking). Is there any reference mentioning the term in news recommendation field?
>
> Thanks for pointing out that, it is an excellent question related to the crucial point of dual-encoder model, as well as the initial enlightenment of this paper, which is no harm to discuss more.
> Using the term "recall" instead of "ranking" because: First, from definition perspective, news recall [2, 3] (or new candidate generation [4] or news retrieve [5]) is mainly to retrieve a small amount news from database.  Secondly, from the usage of dual-encoder models, previous studies [6] and industry practice show that,the models are mostly used to recall news because their high efficiency. On the other hand, Dual-encoder models has its own inherent challenge on weak interaction, which is less competitive comparing with strong interaction models. And almost news ranking models have strong interaction, that is why we use the term "recall" instead of "ranking".
>
> To response your question on "the task conducted in the experiment is simply the news recommendation (ranking)", we would like to point out that, our experiments are the same as previous works [1,2,3]. The evaluation metrics in our experiments are used on both news ranking models and recall models, such as previous work AUC [5], MRR [6], and NDCG [7]. On the purpose of utilizing experimental results from previous works, our experiments apply the same.
>
> [1] Yang Yu, Fangzhao Wu, Chuhan Wu, Jingwei Yi, and Qi Liu. 2022. Tiny-NewsRec: Effective and Efficient PLM-based News Recommendation. EMNLP'22.
>
> [2] Chuhan Wu et al. Two birds with one stone: Unified model learning for both recall and ranking in news recommendation. Findings of ACL'22
>
> [3] Jian Li et al. MINER: Multi-Interest Matching Network for News Recommendation. Findings of ACL'22
>
> [4] Paul Covington et. al. Deep Neural Networks for YouTube Recommendations. RecSys2016
>
> [5] Yuan Wang et. al. An Empirical Study of Selection Bias in Pinterest Ads Retrieval. KDD'23
>
> [6] Omar Khattab et al. ColBERT: Eficient and Effective Passage Search via Contextualized Late Interaction over BERT. SIGIR '20
>
> [7] Yukuo Cen et al. Controllable Multi-Interest Framework for Recommendation. KDD'20

---

### Meta-Review · Area_Chair_wqjC · 2023-09-22

**Recommendation:** 2

**Metareview:**

This submission studies news recommendation, which is an important application of text mining and text retrieval. This submission focuses on dual-encoder architecture and aims to address its two challenges: irrelevant word distraction and weak dual-encoder interaction. Two components are proposed to address these two challenges. However, this submission lacks discussion and comparison with recent work and needs further experiments on datasets beyond MIND such as Addressa (its textual content can be obtained by request).

---

### Decision · Program_Chairs · 2023-10-07

**Decision:**

Accept-Findings

**Comment:**

This submission studies news recommendation, which is an important application of text mining and text retrieval. This submission focuses on dual-encoder architecture and aims to address its two challenges: irrelevant word distraction and weak dual-encoder interaction. Two components are proposed to address these two challenges. However, this submission lacks discussion and comparison with recent work and needs further experiments on datasets beyond MIND such as Addressa (its textual content can be obtained by request).